# Effect of Temperature on the Tribological Properties of Selected Thermoplastic Materials Cooperating with Aluminium Alloy

**DOI:** 10.3390/ma14237318

**Published:** 2021-11-29

**Authors:** Anita Ptak, Paula Taciak, Wojciech Wieleba

**Affiliations:** Department of Fundamentals of Machine Design and Mechatronic Systems, Faculty of Mechanical Engineering, Wroclaw University of Science and Technology, ul. I. Lukasiewicza 5, 50-371 Wroclaw, Poland; paula.taciak1994@gmail.com (P.T.); wojciech.wieleba@pwr.edu.pl (W.W.)

**Keywords:** friction, polymer, low temperature, lubricating oil, CCRD

## Abstract

This article concerns the tribological properties of three selected polymer materials: polyamide PA6, polyethylene PE-HD and polyetheretherketone composite PEEK/BG during sliding against aluminium alloy EN AW-2017A in the presence of hydraulic oil HLP 68. The tests were carried out under contact pressure *p* of 3.5–11 MPa at ambient temperature *T* ranging from −20 °C to +20 °C. The dependence of kinetic friction coefficient *μ_k_* on the two parameters was determined through tribological tests carried out using a pin-on-disc tribometer. A five-level central composite rotatable design (CCRD) was adopted for the experiment. All the test results were statistically analysed. The microhardness of the surface of the polymeric material was measured before and after the friction process. The surface was also examined under SEM. Temperature and contact pressure have been found to have a significant effect on the tribological properties of the tested sliding pairs. Relative to the applied friction conditions, the surfaces after friction showed rather heavy signs of wear.

## 1. Introduction

In recent years there has been a significant interest in the use of polymeric materials for machinery and equipment parts and components. Therefore, much of the relevant tribological research is devoted to the motion properties of the interacting surfaces of sliding pairs [1,2,3,4,5]. These studies focus mainly on the interaction between steel and high-performance polymers, mostly in dry friction conditions and at room temperature. Publications on tribological tests of high-load joints under mixed friction conditions are considerably fewer. However, mixed friction conditions can occur, for example, during the operation of hydraulic system components, such as valves and pumps, used in service vehicles working in temperate climates. The joints in these components are exposed to low temperatures (reaching −30 °C) in winter, which can significantly adversely affect the interaction between the parts, especially if one of the interacting materials is a polymer or a polymer-based composite.

Polymeric materials are increasingly used in machinery and equipment construction because of the following advantages: low price, low density, corrosion resistance and chemical resistance. Unfortunately, the properties of polymers are susceptible to change at ambient temperature and at the temperatures occurring in tribological joints. The influence of temperature changes on the moduli of longitudinal elasticity, thermal expansion and relaxation of the most commonly used thermoplastic polymeric materials has been described in detail in [6,7,8,9,10,11]. According to these studies, the Young’s modulus of the polymeric materials increases as the temperature decreases. The tendency of the changes depends on the type of material. This is due to various thermomechanical states and thermal transitions and to how the properties of polymeric materials change with increasing/decreasing temperature. It is reported that the polymeric materials used in mechanical engineering are almost exclusively between the brittle temperature and a temperature slightly above the glass transition temperature [12]. Significant changes in the mechanical properties of polymeric materials occur in the temperature range of −30 °C to +60 °C, which is the most common operating temperature range of machine components made of polymers [13]. As the temperature falls, thermoplastic polymer materials become harder and brittler, and their impact resilience decreases, as demonstrated in, e.g., study [7] where the mechanical properties of glass fibre reinforced composites (PEEK + 30% GF) were tested in a wide temperature range: from room temperature to almost absolute zero. The test results show that as temperature decreases, the modulus of elasticity, the modulus of elasticity in bending and elongation at break increase, while impact resilience decreases.

Furthermore, stress in polymeric materials has been found to depend on temperature. The author of [14] concludes that for the tested materials (PA6 and amorphous PVE), the lower the temperature, the higher the stress at which the sample breaks. This is connected with the orientation of the amorphous and crystalline phases. The elongation of the sample at break becomes smaller as the temperature falls and disappears almost entirely at *T* = −80 °C (brittle break at deformation amounting to a few per cent). The author highlights the importance of the physical state of the materials during testing. Lower operating temperatures contribute to a reduction in heat conductivity, which has a bearing on friction processes (polymeric materials heat up less). Moreover, the coefficient of thermal expansion is reduced [8,9].

Low temperatures, especially the sub-zero ones, affect the mechanical, thermal, chemical, electrical [15,16,17,18] and tribological properties of polymeric materials [10,19,20]. There are numerous theories on how temperature affects the frictional resistance of such materials. For example, Wang [21], on the basis of his experimental results, concluded that as temperature falls, the coefficient of friction initially decreases and then increases, but is still lower than at room temperature. This is because polymers cannot form a polymeric film on the interacting element at negative temperatures. Hence, not polymer-polymer friction, but polymer-metal friction occurred in the investigated case. As temperature falls, the polymer’s hardness increases, and its deformation decreases, and as a result, the contact area decreases. The wear rate decreases steadily with decreasing temperature. Wieleba et al. [22] found that in the case of thermoplastic polymers and silicone rubber, the static friction coefficient decreases with increasing temperature. This is due to an increase in the thickness of the water layer formed on the ice surface as a result of ice melting. Polymers show poor surface wettability with water. Below −20 °C, the friction coefficient of elastomers decreases, which is attributed to a change in their material properties, i.e., an increase in stiffness. Study [10] showed that as temperature falls from 0 °C to −50 °C, the value of kinetic friction coefficient *µ_k_* decreases. This was ascribed to the presence of condensed water (or frost), acting as a lubricant in the friction area at low temperatures, resulting in weaker frictional resistance despite the increased stiffness of the polymeric material.

Typical structural steels are most commonly used for interaction with polymers. However, the research on non-ferrous metal alloys [18] is worth noting, especially if the joints are to work in contact with water or other chemically aggressive media. In such cases, stainless steel or aluminium alloys are used. Replacing steel with an aluminium alloy offers many advantages, especially the reduced weight of the components at maintained strength properties. For this reason, this material is increasingly used in dry and lubricated sliding contacts [11,23,24,25,26,27,28,29].

Unfortunately, because of the lack of unified procedures and standards, and the variety of parameters used, it is difficult to compare the test results reported in the literature. This was the reason for taking up the topic, which would partially complement the existing knowledge.

This paper deals with the effects of temperature and unit pressure on the coefficient of friction of polymer-aluminium alloy EN AW-2017A sliding pairs. Three thermoplastic polymers were chosen for the study: polyamide PA6, polyethylene PE-HD and polyetheretherketone composite PEEK/BG. The tests were carried out under contact pressure *p* 3.5–11 MPa at ambient temperature *T* from −20 °C to +20 °C in the presence of hydraulic oil HLP 68. SEM and microhardness tests of the friction surfaces complemented the tribological studies.

## 2. Materials and Methods

### 2.1. Samples

Three polymeric materials were investigated: PEEK/BG (polyetheretherketone with 10% PTFE, 10% carbon fibres and 10% graphite), PA6 (polyamide-6) and PE-HD (high-density polyethene) (Table 1). These polymers were chosen not only for their wide application, availability and high chemical resistance, but also for their good sliding properties (a low coefficient of friction) and mechanical properties. The samples were made in the shape of a pin with a Ø4 mm diameter, installed in a holder (Figure 1). They interacted with an EN AW-2017A aluminium alloy disc (Table 2). The friction track radius amounted to 30 mm.

Duralumin EN AW-2017A was used for machine parts and components, automotive industry components and military equipment. Owing to its high strength-to-weight ratio (much higher than that of steel), it is also used for the production of assemblies and structural elements in the aviation industry [30]. Moreover, aluminium alloys have begun to be used in hydraulic devices. Duralumin performs well in mixed friction conditions owing to its high strength and the fact that lubrication results in more effective heat dissipation from the friction zone and substantially reduces the friction force. Duralumin can be used when it is important to reduce the weight of a machine’s superstructure.

The test parameters were selected with regard to applications in heavily loaded machines and devices. Hydraulic oil *HLP 68* (PKN ORLEN SA, Poland) was used in the mixed-friction tests. It is a high-quality mineral oil used in hydrostatic hydraulic systems, heavily loaded power transmission systems and force systems (such as adjustment mechanisms and hydraulic gears). The oil has very good anti-wear properties [31]. The properties of hydraulic oil are presented in Table 3.

### 2.2. Method of Measuring Friction

Tribological tests of polymer-aluminium alloy sliding pairs were conducted on a pin-on-disc test stand to determine the influence of the type of polymer material and the friction process motion parameters, e.g., sliding speed, temperature and surface pressure, on the tested material’s frictional characteristics.

A scheme of the sliding pair is shown in Figure 2a. The tested polymer material in the form of a pin 4 mm in diameter was installed in a holder protecting the sample against deformation during the friction process. The pin was sliding against a disc (a counterface) made of aluminium alloy EN AW-2017A. The normal force *F_N_* acted perpendicularly to the disc. The disc rotated at a constant speed during the test. The polymeric material was sliding on the counterface within the track radius of 30 mm. The tested pairs were placed in a climatic chamber in which temperature could be adjusted from +150 °C to −70 °C and relative humidity from 20% to 98%. Force *F_t_*, called the friction force, was recorded during the rotational motion of the disc in the tests. Photographs of the test stand measurement head inside the chamber are shown in Figure 2b.

Because of the temperatures involved, the friction force sensor must be resistant to both low and high temperatures. Therefore FBG (Fibre Bragg Grating) optic deformation sensors with a modern laser reflection-based measuring method consisting of changing the linear Bragg wavelength were used. FBG sensors are characterised by electric passivity, resistance to external radio frequency sources and resistance to electrostatic interference. They enable highly accurate measurements in the temperature range of −45 °C to +110 °C.

### 2.3. Microhardness and Microscopic Examinations

As the two materials rub against each other during the tribological test, the surface layers undergo changes. Most importantly, the physical structure of the polymeric material is modified as a result of the mechanical, thermal and chemical interactions due to friction. This modification manifests itself mainly in changes in the mechanical properties of the polymeric material’s surface layer [32]. One way to determine these changes is to measure the material’s microhardness before and after the test.

Microhardness was measured using the SHIMADZU HMV-2T tester (SHIMADZU, Germany). The measurement consisted of pressing the Knoop diamond indenter [33] with force F = 980.7 mN at dwell time t = 30 s (HV0.1) into the polymeric material at defined measuring points. This method was chosen because of the small indentation depth, whereby the microhardness of thin surface layers could be determined. The dwell time and load were chosen to obtain a durable indentation in the tested materials. Because of the viscoelastic properties of polymers, the size of the indentation would change quickly when a too short dwell time was adopted, which would result in large measurement errors. Measurements were performed four times at measuring points evenly distributed on the surface (Figure 3).

The surface of the polymeric material needs to be examined to determine the friction mechanism, which occurs in heavily loaded sliding pairs at low temperatures. Changes in the surface are usually assessed through microscopic examinations, and this method was employed in this study. The Phenom-World ProX scanning electron microscope (ThermoFisher Scientific, Waltham, MA, USA) was used for this purpose.

### 2.4. Central Composite Rotatable Design of Experiment

Since it is problematic to select proper measuring points for the experiment, the central composite rotatable experimental design was adopted for two variables: ambient temperature and contact pressure. The rotatable design is suitable owing to the constancy of the regression function in the vicinity of the central point. The obtained second-degree polynomials form the regression function of the measured quantities, as in Formula (1).
(1)$${y=a}_{0\;}{+a}_{1\;}x_{1\;}{+a}_{2\;}x_{2\;}{+a}_{3\;}x_{1}^{2}{+a}_{4\;}x_{2}^{2}{+a}_{5\;}x_{1\;}x_{2}$$
where: a_0_, a_1_, …, a_5_—polynomial coefficients, x_1_, x_2_—two variable quantities (unit pressure, temperature).

The coefficients were calculated using the least-squares method. This is the oldest and most important method used in statistics to determine linear regression and a trend line for a set of data. In this way, the most optimal line (a linear model) representing the relationship between x and y can be found. The least-squares method can also be used to determine nonlinear relationships.

The regression functions constituted the basis for creating the kinetic friction coefficient’s contour and its spatial graphs as a function of two variables (*p*, *T*). In addition, the functions were subjected to a statistical analysis, which consisted of calculating the standard deviation, *F*-test and correlation coefficient *R*. This provided the basis for the statistical verification of the test results.

The ranges of the adopted variables were: contact pressure *p* = (3.5–11 MPa), and temperature *T* = (−20–20 °C). The ranges reflect the environmental conditions prevailing in temperate climates. All tests were performed at a constant sliding speed of 0.5 m/s. The basic test parameters are presented in Table 4. On the basis of the ranges of contact pressure and temperature, 13 measuring points (including five-fold measurement in the central point) were selected for the experiment (Figure 4).

## 3. Results and Discussion

The results of the tribological tests were the basis for generating regression functions as described in Section 2.4. The functions were used to determine surface and contour characteristics. In this way, the dependence of kinetic friction coefficient *µ_k_* on temperature T and contact pressure p could be better illustrated.

The tribological tests were supplemented with investigations aimed at determining the surface layer condition. Before and after the friction process, the microhardness of the polymer surface layer was measured using the Knoop method. After the tribological tests, the sliding surfaces of the tested polymers were studied under a scanning electron microscope (SEM).

### 3.1. Effect of Temperature and Contact Pressure on Coefficient of Friction of Polymer-Aluminium Alloy Sliding Pair

The results of the tribological tests are presented as surface contour diagrams in Figure 5, Figure 6 and Figure 7. The regression functions and their statistical evaluation can be found in Table 5. The statistical analysis validated the regression functions used to describe the test results.

In the case of PEEK/BG, the lowest coefficient of kinetic friction (*μ_k_* = 0.05) occurs at the contact pressure of 5.75–9.5 MPa and the temperature of −2 °C to 10 °C. The highest coefficient of kinetic friction for PEEK/BG (*μ_k_* = 0.19) occurs at the lowest temperature and the lowest contact pressure. As contact pressure increases and temperature decreases, the coefficient of friction initially decreases to a minimum and subsequently increases. The change in the coefficient of friction has the same character for both contact pressure and temperature, but differs in its intensity.

A similar dependence was observed for PA6, but the lowest kinetic friction coefficient values (*μ_k_* = 0.11) were registered in the temperature range of 2–6 °C at the contact pressure of 7–11 MPa. The highest kinetic friction coefficient (*μ_k_* = 0.17) occurred at the lowest contact pressure (*p* = 3.5 MPa) and the lowest temperature (*T* = −20 °C).

PE-HD shows a completely different pattern of the dependence of the coefficient of friction on the two analysed parameters. At the lowest contact pressure, as the temperature decreased, the kinetic friction coefficient increased steadily at a high rate (from *μ_k_* = 0.04—the function minimum to *μ_k_* = 0.18—the function maximum). The pattern of the friction coefficient-temperature dependence at the highest unit pressure was comparable with that observed for PEEK and PA6. When temperature decreased, the coefficient of friction initially decreased to a minimum and then increased.

### 3.2. Change in Microhardness after Friction Process

Microhardness tests were carried out before the friction process, to determine the microhardness of the initial surface layer of the polymer, and after the friction process, to determine the changes in the properties of this surface layer.

Table 6 presents the results of the microhardness tests and their statistical interpretation in the form of the arithmetic mean, standard deviation and 95% confidence interval. The data are also presented in the form of a graph (Figure 8).

Figure 8 shows a significant difference between the initial and final values of microhardness. Hence, it can be concluded that the friction process significantly affects the hardness of the surface layer of the polymeric materials. The microhardness of the surface layer of PE-HD and PEEK/BG increased after the friction process, whereas that of PA6 decreased. The changes expressed in percentages are as follows:PEEK/BG—an increase of 12.2%;PA6—a decrease of 29.3%;PE-HD—an increase of 41.8%.

The indentations in PE-HD and PA6 were very narrow, and wear scratches were visible on the surface of the polymers. Consequently, it was difficult to determine the length of the analysed diagonal, and a detailed assessment of the friction surfaces had to be made.

### 3.3. Analysis of Sliding Surface of Tested Polymeric Materials

Selected SEM micrographs of the sliding surface of the polymeric materials are shown in Figure 9, Figure 10 and Figure 11.

On the basis of the micrographs, inferences about the wear processes could be drawn. The scratches and other traces of wear clearly visible on the surface of the tested samples indicated that abrasive wear occurred despite the presence of a lubricant in the mixed friction regime. There were scratches and grooves on all of the examined surfaces. In the case of PA6 (Figure 10c) and PE-HD (Figure 11b), the grooves were larger and had irregular, curved edges. In the case of PEEK/BG (Figure 9a), they were shallow with small edge folds. Moreover, they were significantly fewer than in the case of the other materials. The occurrence of grooves indicates an increased friction temperature and the plasticisation of the surface of the materials. PEEK/BG is a composite reinforced with PTFE, graphite and carbon fibres, which may, to some extent, reduce wear.

All the scratches and grooves were aligned in the same direction. Numerous wear products (debris) pressed into the polymer surface could be seen in the grooves. The debris was small in size, but in some places, it was larger. Its size was 120 µm for PEEK/BG (Figure 9c), 40 µm for PE-HD (Figure 11a) and 70 µm for PA6 (Figure 10a). The presence of debris is probably due to the detachment of the polymer matrix from the surface and its hardening at the low temperature prevailing outside the friction area. When this wear product gets between the metal surface and the plasticised polymer surface, it is pressed into the latter.

The smallest wear products with larger dimensions were found in the case of PE-HD. This may be due to the different physical state of this polymer during the tribological tests. PE-HD was in a viscoelastic state (*T_g_* = −110 °C), whereas the other two materials were in a glassy state. In the viscoelastic state, adhesive interactions are equally common as mechanical interactions [13].

There are characteristic long fragments of pressed-in material of similar shapes along their entire length on the surface of PEEK/BG and PA6 (Figure 9b and Figure 10b). They were probably rolled between the hard surfaces of the polymer and the aluminium alloy or resulted from material residues being drawn from the sample’s edge into the contact zone. Some of the material was torn off and immediately pressed into the polymer’s surface (Figure 9b, Figure 10a and Figure 11c). This effect, together with the wear products, significantly affects surface roughness. No evidence of degradation, which could be expected at the low temperature (*T* = −20 °C), was found on the tested surfaces. This could be due to the rise in frictional temperature, being the response of the tribological system.

### 3.4. Summary of Results

Three polymeric materials: PEEK/BG, PA6 and PE-HD frictionally interacting with aluminium alloy EN AW-2017A in mixed friction conditions, were tested. The results of the tribological tests show that the kinetic friction coefficient of the selected materials depends on temperature and contact pressure. Moreover, the polymeric material’s microhardness and surface condition were determined before and after the tribological test. The following observations can be drawn from the study:Temperature and contact pressure significantly affect the kinetic friction coefficient, and the friction characteristics are not stable over the entire range of the parameters.The friction characteristics of PE-HD are different from those of PEEK/BG and PA6. When temperature decreases from 0 °C to −20 °C, the increase in the coefficient of friction can be due to both an increase in the rigidity of the polymeric materials and an increase in the viscosity of the lubricant. The increase in the rigidity of the polymers reduces their deformation, while the increase in the viscosity of the oil causes greater resistance of the liquid layers moving relative to each other.Regarding changes in the kinetic friction coefficient, PA6 can be considered the most stable of all the studied materials. The kinetic friction coefficient (*µ_k_*) of PA6 ranges from 0.11 to 0.17, that of PEEK/BG from 0.05 to 0.19 and that of PE-HD from 0.04 to 0.18.The highest kinetic friction coefficient for all the tested materials occurred at the lowest temperature (*T* = −20 °C) and the lowest unit pressure (*p* = 3.5 MPa). The coefficient of kinetic friction decreased with increasing contact pressure, which can be due to the fact that a greater amount of heat is generated in the sliding pairs during friction. This is confirmed by the condition of the abrasive surfaces of the analysed materials. Numerous furrows were visible, which is evidence of the plasticisation and heating up of the polymer material. In the case of PEEK/BG, the furrows were fewer and shallower, which can be due to this polymer’s composite structure reinforced with carbon and graphite fibres.There was numerous debris on the examined surfaces of the polymers. A large amount of debris can indicate high material wear of the material. The debris is generally small in size, but in some locations, larger debris was observed. Its size amounted up to 120 µm in the case of PEEK/BG and up to 70 µm in the case of PA6. There were also over 280 µm long narrow fragments of harder polymer material pressed into the surface. In the case of PE-HD, there was very little debris, and its size did not exceed 30 µm. It can be concluded that in certain conditions, the structure of the surface layer and the behaviour of this material were affected by the viscoelastic state in which it remained at a low temperature.Tribological processes significantly affect the microhardness of the surface layer of polymers. It can be concluded that the changes in microhardness depend on the type of polymer and depend on the changes in the microstructure occurring during friction. In the case of PEEK/BG, it is possible that the measurements were carried out in places where there were fillers increasing microhardness there.

## 4. Conclusions

The following conclusions, valid for the test conditions used in this study, are drawn from the research:The change in operating temperature affects the tribological characteristics of the polymer-aluminium alloy sliding pairs. This can be due to the different physical states of the materials (PE-HD was in a viscoelastic state, while PEEK/BG and PA6 were in a glassy state) in the analysed temperature range [12].The obtained characteristics of the kinetic friction coefficient suggest that PA6 turned out to be the most stable and predictable material in terms of tribological applications.SEM images allow a qualitative study of the wear process. On their basis, one can assume that at low temperatures, the detached polymeric material cures, forms and then gets between the rubbing surfaces, where it is pressed into the more plasticised surface of the polymer.Tribological processes significantly affect the microhardness of the surface layer of polymers. It is related to the change in the properties of thermoplastic polymers during the temperature reduction [7,12]. Decreasing the temperature causes an increase in the stiffness of polymeric materials (at the same time, a decrease in their deformation) and a change in the ordering of the structure of thermoplastic semi-crystalline materials [14].

Tribological tests supply much information about the behaviour of polymeric materials in specific conditions. In the case of highly loaded mechanisms, especially friction joints, it would be worth supplementing such tests with investigations of the wear of polymeric materials at low temperatures and with analyses of the wear products.

## Figures and Tables

**Figure 1 materials-14-07318-f001:**
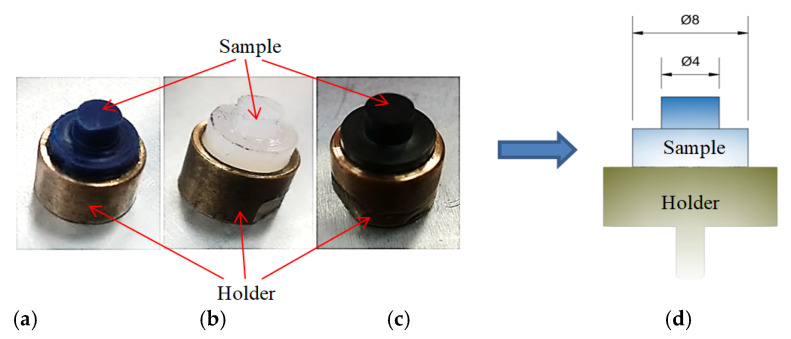
Samples of tested materials: (**a**) PE-HD, (**b**) PA6, (**c**) PEEK/BG, (**d**) scheme of polymer sample placed in holder.

**Figure 2 materials-14-07318-f002:**
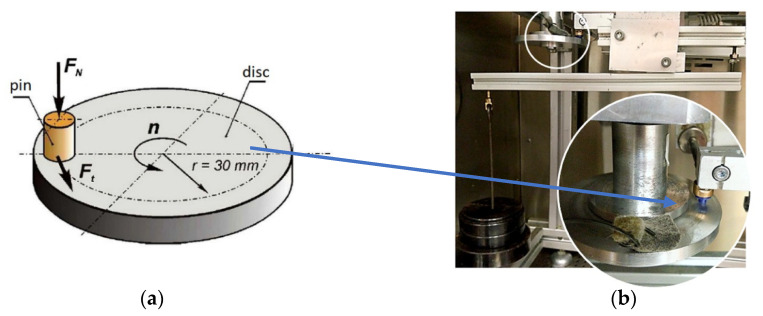
Pin-on-disc sliding pair: (**a**) scheme, (**b**) photos of test stand measuring head inside the chamber.

**Figure 3 materials-14-07318-f003:**
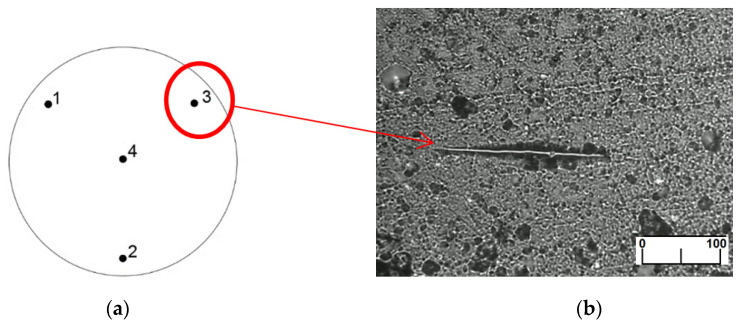
Microhardness measurement: (**a**) distribution of measuring points on surface of tested sample, (**b**) exemplary indentation in surface of polymeric material (PEEK/BG).

**Figure 4 materials-14-07318-f004:**
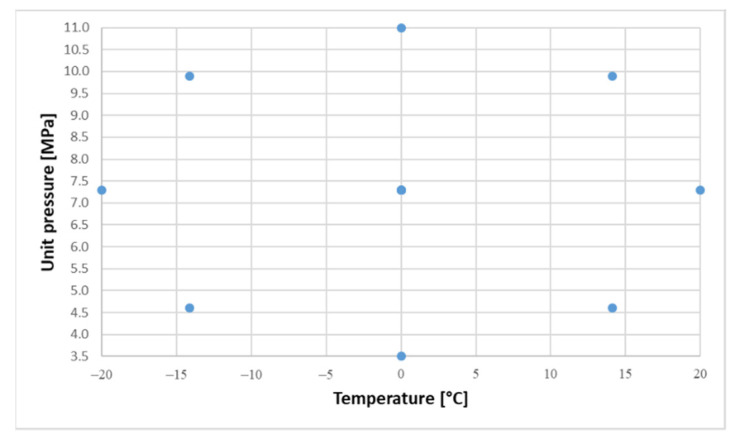
Distribution of measuring points for two variables (*T*, *p*).

**Figure 5 materials-14-07318-f005:**
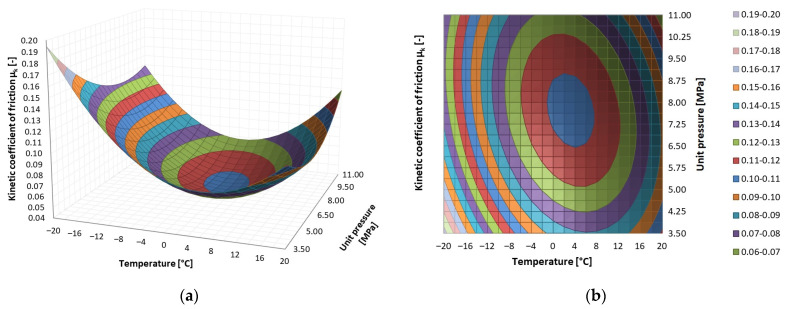
Dependence of kinetic friction coefficient *μ_k_* on temperature and contact pressure for PEEK/BG: (**a**) surface graph, (**b**) area graph.

**Figure 6 materials-14-07318-f006:**
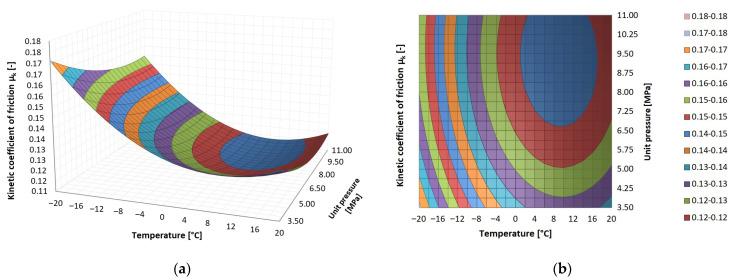
Dependence of kinetic friction coefficient *μ_k_* on temperature and contact pressure for PA6: (**a**) surface graph, (**b**) area graph.

**Figure 7 materials-14-07318-f007:**
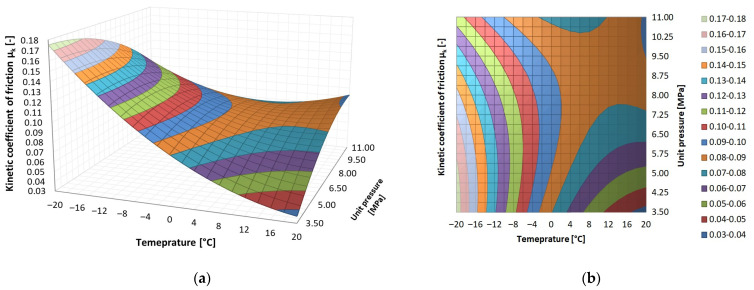
Dependence of kinetic friction coefficient *μ_k_* on temperature and contact pressure for PE-HD: (**a**) surface graph, (**b**) area graph.

**Figure 8 materials-14-07318-f008:**
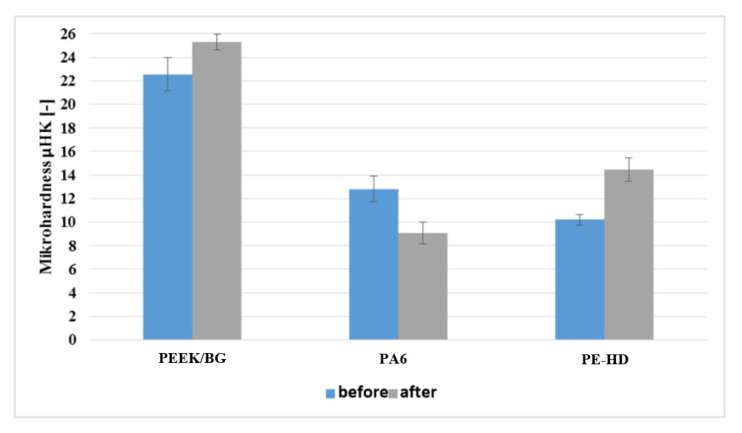
Polymeric material surface microhardness of PEEK/BG, PA6, and PE-HD before and after friction process.

**Figure 9 materials-14-07318-f009:**
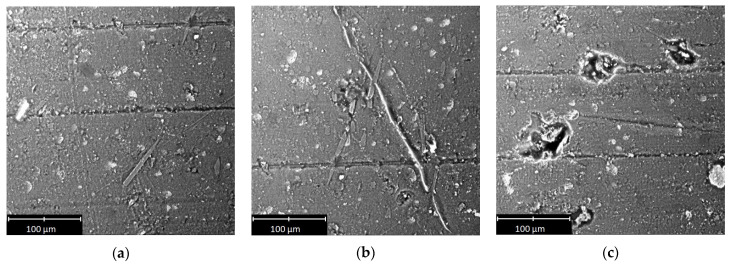
Selected SEM micrographs of sliding surface of PEEK/BG.

**Figure 10 materials-14-07318-f010:**
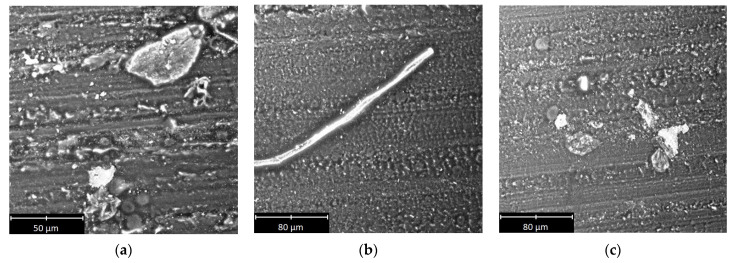
Selected SEM micrographs of sliding surface of PA6.

**Figure 11 materials-14-07318-f011:**
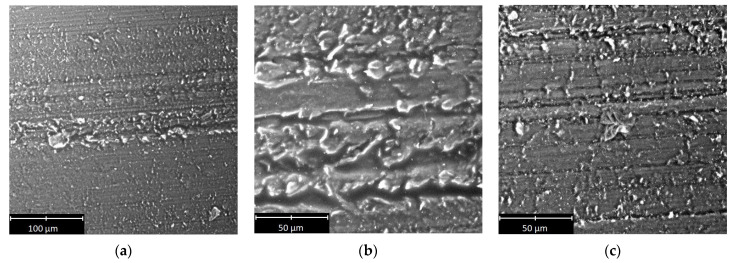
Selected SEM micrographs of sliding surface of PE-HD.

**Table 1 materials-14-07318-t001:** Properties of polymeric materials.

Properties	PEEK/BG	PA6	PE-HD
Composition	10% PTFE 10% carbon fibres 10% graphite	non-modified	non-modified
Flexural modulus	8.10 GPa	2.70 GPa	1.38 GPa
Tensile modulus	10.0 GPa	1.80 GPa	1.55 GPa
Melting point	334 °C	235 °C	135 °C
Glass Transition Temperature *T_g_*	+140 °C	+60 °C	−90 °C
Elongation at Break	3%	260%	600%

**Table 2 materials-14-07318-t002:** Chemical composition and properties of aluminium disc EN AW-2017A.

Chemical Composition									
Mg [%]	Mn [%]	Fe [%]	Cu [%]	Si [%]	Zr + Ti [%]	Cr [%]	Zn [%]	Other	Al [%]
0.4–1	0.4–1	≤0.7	3.5–4.5	0.2–0.8	≤0.25	≤0.1	≤0.25	≤0.05	Rest
**Properties**									
Density					2.79 g/cm^3^				
Modulus of transverse elasticity *G*					27.2 GPa				
Modulus of elasticity *E*					72.5 GPa				
Poisson’s number					0.33				
Pour point					645 °C				
Freezing point					510 °C				
Thermal conductivity					134 W/mK				
Specific heat					873 J/kgK				
Strength properties					very good tensile strength, fatigue strength				
Workability					good				
Hardness					high value (110 HB)				
Resistance of wear					excellent				

**Table 3 materials-14-07318-t003:** The basic properties of hydraulic oil *HLP 68*.

Properties	
Viscosity index	99
Pour point	−30 °C
Flashpoint	228 °C
Kinematic viscosity at 40 °C	66.2 mm^2^/s
Corrosion action on copper plates (100 °C/3 h)	1a degree of corrosion
Deemulsibility, time to oil/water emulsion separation: 40–43 mL of oil 37–40 mL of water 0–3 mL of emulsion	25 min. at 54 °C
Ability to release air at 50 °C	8 min.
Ability to transfer loads with the FZG, breaking load, minimum	10

**Table 4 materials-14-07318-t004:** Tribological test parameters.

Experimental Parameters	
Polymer materials (pin)	PE-HD
	PA6
	PEEK/BG
Metal disc (counterface)	aluminium alloy EN AW-2017A
Range of unit pressure *p*	3.5–11 MPa
Range of temperature *T*	−20–20 °C
Constant sliding speed	0.5 m/s
Sliding distance (for one measurement)	0.3 km
Environment	Mixed friction—HLP 68 hydraulic oil in place of contact

**Table 5 materials-14-07318-t005:** Polynomial coefficients and statistical evaluation of regression functions for tested polymers.

Polynomial Coefficients	Tested Polymers		
	PEEK/BG	PA6	PE-HD
a_0_	0.172163019	0.15627976	0.0414
a_1_	−0.002913008	−0.00102	−0.0048
a_2_	−0.031226608	−0.008566	0.01485938
a_3_	0.000165563	4.96875 × 10^−5^	0.000063125
a_4_	0.001971556	0.000453333	−0.001048889
a_5_	0.00024	1.33333 × 10^−5^	0.000386667
**Statistical Evaluation**			
Standard deviation	0.0313	0.0146	0.0287
Correlation coefficient R	0.9751	0.9509	0.9745
(for a = 0.01, Rkr = 0.886)	27.12	13.21	14.35

**Table 6 materials-14-07318-t006:** Microhardness test results and their statistical interpretation.

Measuring Point	Microhardness *µHK*					
	PEEK/BG		PA6		PE-HD	
	before	after	before	after	before	after
1	21.5206	25.881	13.677	8.76147	10.5266	15.0473
2	23.6849	24.8916	12.9428	9.30893	9.98847	14.4064
3	22.4294	25.2077	12.5415	8.44347	9.90936	14.7573
4	22.5639	25.2077	12.1053	9.75392	10.3565	13.6135
Arithmetic mean	22.5497	25.297	12.8167	9.06695	10.1952	14.4561
Standard deviation	0.88739	0.41687	0.66779	0.58096	0.29455	0.61986
Confidence interval	1.41204	0.66334	1.06261	0.92443	0.4687	0.98634

## Data Availability

The data presented in this study are available on request from the corresponding author.

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
