# Peer review of "Effect of Temperature on the Tribological Properties of Selected Thermoplastic Materials Cooperating with Aluminium Alloy"

_materials, 2021, doi:10.3390/ma14237318_

Round 1

Reviewer 1 Report

The author has answered the previous review comments, but there is still a key question to explain.
In the experimental design, the author did not consider the influence of hydraulic oil on polymer materials in the friction process. In fact, rubber compatibility is a key index in the performance of hydraulic oil, and different oil products will also have a great impact on the hardness of polymer. This leads to the results of this paper may be caused by many factors, not just friction test.

Reviewer 2 Report

The reviewer thanks the authors for the submission and the Editor for the invitation to review it.

In the proposed manuscript, the authors investigated experimentally the impact of the contact pressure and the temperature on the tribological behavior of three polymer materials. In this perspective, tribological tests of polymer-duralumin are performed on a pin-on-disc test under lubrification with HLP68 hydraulic oil. The results provided by the authors shows the influence of the type of polymer material and the mixed-conditions (temperature, contact pressure) of the friction process on the kinematic friction coefficient and the microhardness of the studied polymers.

The reviewer has few concerns about the manuscript in its current form and he recommends its publication in Materials after minor revision denoted below.

Minor revisions and questions

  1. At the end of the introduction, the reviewer suggests the authors to add a paragraph reminding the main objectives of the study and introducing the plan of the article.
  2. The last sentence of the introduction indicates that it is difficult to compare the test results reported in the literature due to a lack of unified procedures and the variety of parameters used. To what extent does the work reported in this article answer to this issue?
  3. Table 2: Please convert the two moduli of aluminum disc in GPa, it will be more consistent with the unit used for the moduli of polymeric materials given Table 1.
  4. line 155: Remove the word "their' in the sentence "As the two materials (...) tribological test their the surface ..."
  5. line 200-201: The authors specify that 13 points were selected to perform the measurements, refering on Figure 4. However, this figure depicts only 9 points. Can the authors clarify this misunderstanding?
  6. Section "Results": On the whole, the reviewer appreciates the accurate description of Figures. However, he also thinks that the results should be discussed in this section and be interpreted thanks to previous results coming from literature and working assumptions. 
  7. Section 3.1: The authors have chosen to describe Figures 5-7, beginning with the case of PEEK/BG, continuing with PA6 one and finishing with PE-HD case. So, the reviewer rather suggests to associate the Figure 5 with the case of PEEK/BG, Figure 6 with the PA6 one and Figure 7 with the PE-HD case, in order to be more consistent with the order chosen by the authors for the description.
  8. Section 3.2: For the same reason as previously, please reorder the columns of Table 6 as follows: PEEK/BG, PA6 and PE-HD.
  9. Section 3.3: SEM micrographs of the sliding surface of the three studied polymer materials are shown in Figure 9-11. In which area of the pin are made the observations? The reader has no information concerning the differences between the Figures (a); (b) and (c) for the same material: different observation areas? Different mixed-conditions (temperature, pressure)? It is necessary to precise the conditions of the experiments in order to then provide a sound analysis of the results.
  10. Section 3.3: No description of Figures 10(c) and 11(b) are provided in the text, while the other ones are explained in details. Maybe an oversight?
  11. Conclusions: The discussions of the results are given here. As suggests previously by the reviewer, maybe it would preferable to provide this analysis of th results in the previous section of the manuscript.
